# *DySR*: ADAPTIVE SUPER-RESOLUTION VIA ALGORITHM AND SYSTEM CO-DESIGN

**Syed Zawad**[1*]**, Cheng Li**[2]**, Zhewei Yao**[2]**, Elton Zheng**[2]**, Yuxiong He**[2]**, Feng Yan**[3]
[1]University of Nevada, Reno, [2]Microsoft Research, [3]University of Houston
szawad@nevada.unr.edu,
{chengli1,zheweiyao,elton.zheng,yuxhe}@microsoft.com,
fyan5@central.uh.edu

## ABSTRACT

Super resolution (SR) is a promising approach for improving the quality of low resolution streaming services on mobile devices. On mobile devices, the available computing and memory resources change dynamically depending on other running applications. Due to the high computation and memory demands of SR models, it is essential to adapt the model according to available resources to harvest the best possible model performance while maintaining quality of service (QoS), such as meeting a minimum frame rate and avoiding interruptions. Nevertheless, there is no SR model or machine learning system that supports adaptive SR, and enabling adaptive SR model on mobile devices is challenging because adapting model can cause significant frame rate drop or even service interruption. To address this challenge, we take an algorithm and system co-design approach and propose a *Dynamic Super Resolution* framework called *DySR* that maintains QoS while maximizing the model performance. During the training stage, *DySR* employs an adaptation-aware one-shot Neural Architecture Search to produce sub-graphs that share kernel operation weights for low model adaptation overhead while striking a balance between performance and frame rate. During the inference stage, an incremental model adaptation method is developed for further reducing the model adaptation overhead. We evaluate on a diverse set of hardware and datasets to show that *DySR* can generate models close to the Pareto frontier while maintaining a steady frame rate throughput with a memory footprint of around 40% less compared to the assembled baseline methods.

## 1 INTRODUCTION

Deep super-resolution (SR) has been widely used in applications such as medical imaging (Li et al. (2021)), satellite imaging (Shermeyer & Van Etten (2019)), and image restoration (Qiu et al. (2019)). SR has attracted lots of attentions in recent years due to the surging demands in mobile services such as video conference, content sharing, and video streaming, where it helps provide high-resolution visual content from low-resolution data source (Zhang et al. (2020); Li et al. (2020; 2021)). SR models are resource demanding (Li et al. (2021); Lu & Hu (2022)) and need to meet Quality of Service (QoS) standards to provide good user experience in visual services. Examples of QoS including meeting a minimum frame rate and avoiding interruptions so that users perceive smooth motions. This, however, is challenging for mobile devices where computing and memory resources are limited and the availability of which also depends on other running applications. To meet QoS for different mobile devices, existing works develop models for specific devices (Liu et al. (2021b); Lee et al. (2019); Ayazoglu (2021)) or use Neural Architecture Search (NAS) (Chu et al. (2021); Guo et al. (2020); Huang et al. (2021)) to generate multiple hardware-tailored models. However, none of these approaches considers the fluctuating resource environment of mobile devices and often leads to poor QoS. One potential way to achieve good QoS is to dynamically adapt the model based on available resources. The challenges are two folds. First, how to design an adaptive model. Second, how to enable model adaptation in a live inference system.

---

*Work done in part as an intern with the Microsoft DeepSpeed team.

To enable adaptive model, we employ NAS to generate a set of models with different sizes so that the most profitable model is used under each resource availability situation to ensure a steady frame rate while maximizing the model performance. Unfortunately, none of existing machine learning frameworks supports live model adaptation. To enable model adaptation in actual system, we explore two ideas. The first idea is to use an assemble method to keep all models loaded in the system at all times to avoid model switching overhead. However, such a method results in a significantly larger memory footprint, which is unsuitable for mobile devices. The second idea is to load a single model at a time, but the the model switching overhead is high as it interrupts the steaming for 1-3 seconds each time it switches models, leading to even worse QoS.

To achieve low resource consumption while minimizing the model switching overhead, we propose *DySR* [1], an algorithm and system co-design approach for adaptive SR. To keep a small memory footprint and minimize adaptation overhead, *DySR* employs an adaptation-aware one-shot NAS approach, where a large meta-graph is trained in one-shot such that sub-graphs share kernel operation weights while exploring the best tradeoffs between performance and frames-per-second (FPS). During inference, the meta-graph is fully loaded in the memory and operations are dynamically adapted according to the real-time resource availability in an incremental manner, i.e., only affected operations are swapped or rerouted. Since we do not need to load new models from the hard disk, there is no data transfer overhead.

We evaluate *DySR* against baselines across a wide variety of hardware (from powerful GPUs to low-end mobile processors) using image and video SR datasets (e.g., Urban100 (Huang et al. (2015)) and Vimeo90k (Xue et al. (2019))). Results show that *DySR* can generate models close to the Pareto frontier of the performance vs. FPS tradeoffs while maintaining a steady frame rate throughput with low memory footprint (40% less compared to ensemble method).

## 2 RELATED WORKS

**SR.** (Dong et al. (2014)) is among the first works that employs deep learning models for super-resolution. Since then deeper and more complex models such as (Soh et al. (2019); Nazeri et al. (2019)) were proposed for better performance. Generative Adversarial Networks (GANs) (Creswell et al. (2018); Wang et al. (2019); Ahn et al. (2018); Wang et al. (2018a)) and its variations (Prajapati et al. (2021); Guo et al. (2020); Shahsavari et al. (2021)) have been shown to be highly effective in tackling this task. Attention mechanisms were introduced to SR as well (Zhao et al. (2020); Mei et al. (2021); Chen et al. (2021)). Methods such as network pruning, knowledge distillation, and quantization have been applied to reduce computational overhead of existing SR deep learning models (Jiang et al. (2021); Zhang et al. (2021b); Hong et al. (2022); Wang et al. (2021)). However, all the above efforts focus on building a single model for each hardware and do not consider the dynamic resource environment in mobile devices, and thus fall short in meeting streaming QoS for mobile devices.

**NAS**. Earlier neural architecture search methods rely on Reinforcement Learning (RL) (Zoph & Le (2016)) and evolutionary algorithms (Lu et al. (2018); van Wyk & Bosman (2019)) for architecture engineering. However, these methods are extremely resource demanding and often require thousands of GPU hours. Later works such as (Wen et al. (2020); Jin et al. (2019)) introduce performance prediction, shared weight training, and proxy training to speed up the architecture engineering process. DARTS (Liu et al. (2018)) and its followups (Chen et al. (2019); Wu et al. (2019)) adopt a differentiable architecture search paradigm. The once-for-all work (Cai et al. (2019)) proposes the idea of generating a single model for multiple hardware deployments though pruning and model swapping is needed for each deployment. One-shot NAS (Bender et al. (2018)) and its variations (Huang & Chu (2021); Zhang et al. (2021a); Zhao et al. (2021)) can generate models with few search iterations and have been explored for SR (Chu et al. (2021); Guo et al. (2020); Liu et al. (2021a); Zhan et al. (2021)) but existing works only focus on designing a single model and do not consider QoS for streaming on mobile devices. In Section 5, we compare *DySR* with existing SR models. The results show that our model achieves Pareto optimal performance while meeting QoS.

## 3 MOTIVATIONS AND CHALLENGES

On mobile devices, SR tasks are often running along with other applications, e.g., users often watch streaming videos while browsing websites or doing online shopping; making video calls while play-

---

[1]https://github.com/syed-zawad/srnas

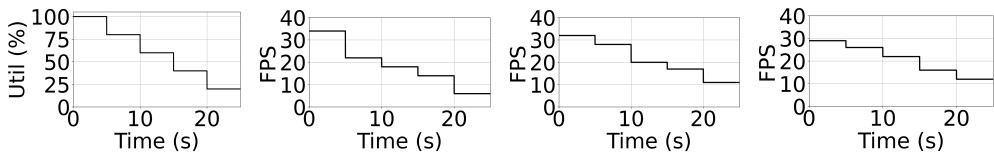

(a) Utilization Trace    (b) Snapdragon 855    (c) Intel i5-560M    (d) 1080Ti

models respectively due to being designed specifically for the corresponding hardware.

Figure 1: (a) Resource utilization cap over time for each device. (b-d) FPS drop due to reduced resources. Uses FALSR-C, FALSR-B (Chu et al. (2021)), and CARN (Ahn et al. (2018))

ing games or taking notes. Therefore, the available computing and memory resources for SR is constantly changing on the already resource constraint mobile devices. We demonstrate the impact of changing available resources on frame rate for static models running on real mobile devices. We test a set of state-of-the-art SR models on the mobile devices they were targeted for and manually limit the amount of resource available over time, i.e., by limiting GPU power and adding background loads to CPU. We use a workload trace to show how the utilization changes over time, as shown in Figure 1. We observe that reducing the processing power available for the models over time results in significant frames-per-second (FPS) drop. This drop at higher utilizations show that the QoS is impacted. Therefore, the state-of-the-art SR models do not perform as expected under practical circumstances, and to maintain QoS (i.e., minimum FPS), models need to be adapted according to the available resources. To fill this gap, however, is challenging. First, we need to design a set of models that can achieve Pareto-optimal performance under different frame rates. However, no existing works can generate such a set of models. Second, we need to adapt models within milliseconds in the real-time system to meet QoS requirements. Unfortunately, no existing machine learning frameworks support real-time model adaptation. One potential solution is to switch between a list of

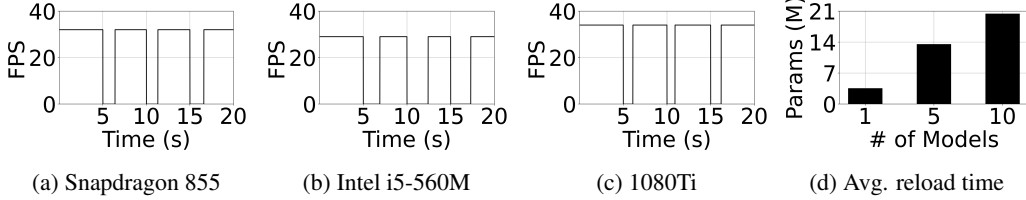

(a) Snapdragon 855    (b) Intel i5-560M    (c) 1080Ti    (d) Avg. reload time

Figure 2: (a-c) FPS over time when models are reloaded every 5 seconds. (d) Total number of parameters in memory against the number of static models loaded as one assembled set.

static models according to the available resources. We prototype this idea and perform experiments on different devices, as shown in Fig. 2. Here, we unload and reload static models as soon as the utilizations change at the intervals 5, 10, and 15 seconds, as shown in (a), (b), and (c) respectively. Across all devices, we observe an interruption of service during each model swapping period. As pointed out in (Liao et al. (2019)), such interruption is the time it takes to read the new model from storage and initialize all the parameters in memory. Pre-loading multiple static models as one is a potential solution to avoid such interruption. However, the assembled model has a significantly higher memory consumption, as shown in Fig. 2d, which is not practical for resource constraint mobile devices.

## 4 ALGORITHM AND SYSTEM CO-DESIGN: *DySR*

To address the aforementioned challenges, in this section, we adopt an algorithm and system co-design approach and propose *DySR*. *DySR* employs adaptive-aware one-shot NAS to create a set of models that achieve Pareto-optimal performance under different frame rate with minimum adaptation overhead during runtime. *DySR* also introduces an incremental adaptation approach to further reduce the model adaptation overhead to meet QoS.

### 4.1 ADAPTATION-AWARE ONE-SHOT NEURAL ARCHITECTURE SEARCH

Neural Architecture Search (NAS) allows creating multiple models in an automated fashion, and thus we choose it for model generation. However, for SR, existing NAS methods only target for a finding a single model with the highest performance while ignoring the frame rate and model adaptation overhand. To provide QoS on mobile devices, we design an adaptive-aware one-shot

NAS method for generating a set of models that achieves Pareto-optimal performance with frame rate constraint and low model adaptation overhead under different hardware devices and resource availability. Following (Liu et al. (2018); Bender et al. (2018)), we define the architecture search space as $S$, which is represented as a single Directed Acyclic Graph (DAG) containing all possible path and operation combinations, called the meta-graph. The main idea here is to train a meta-graph's sub-graph models with Pareto-optimal performance across a range of frame rates. The sub-graphs are all part of the same meta-graph, so that the full meta-graph is deployed and the sub-graphs can be adapted in real time during inference to maintain QoS using an adaptation policy.

***Adaptation-aware Sub-graph Selection***. The generation of adaptation-aware sub-graphs requires two unique criteria to be fulfilled during the NAS search phase to be effective. First, in order to reduce resource consumption, we need to reduce total memory of the meta-graph to be significantly less than an assembled model. Second, we need to minimize the sub-graph adaptation time to ensure uninterrupted service. In order to keep memory consumption at a minimum, we design our meta-graph space to share operations between sub-graphs wherever possible (e.g. two sub-graphs requiring a 3x3x64 convolutional operation at the same layer will use the same meta-graph operation, and thus have intersecting execution paths). To keep the switching time between models low, we need to reduce the number of execution path re-routings between sub-graphs to a minimum. Luckily, we find that both of these properties are related based on the observation that *the number of total operations meta-graph and the number of path re-routes are inversely proportional to the number of shared operations*. In other words, the more operations that are shared among sub-graphs, the less number of redundant operations there are and so meta-graph memory size is reduced. At the same time, more shared operations also means more common execution paths between sub-graphs and so less number of paths need to be changed while adapting.

We use this observation to develop the custom search policy to generate our adaptive models. Thus the objective of our sub-graph selection policy can be formally defined as the constrained joint optimization problem -

$$P^* = \arg\max \; Eval(A(\alpha, D)) + \mu(\alpha, A - \alpha) \; s.t. \; fps(\alpha, D) \geq F_{min}^{\rho} \tag{1}$$

Here, $P$ is the HR quality metric (i.e. PSNR or SSIM), $Eval(A(\alpha, D))$ is the PSNR value of SR output of sub-graph $\alpha$ of meta-graph $A$ on dataset $D$, $\mu$ is the number of operations $\alpha$ has shared with all other sub-graph $A - \alpha$. $fps$ is the frames-per-second and $F_{min}^{\rho}$ is the minimum FPS allowed under available resource $\rho$. Note that $\rho$ can change over time and can be generalizable to any type of computational resource even though in our case we mainly demonstrate using the available processing power. The implementation details are described in detail below.

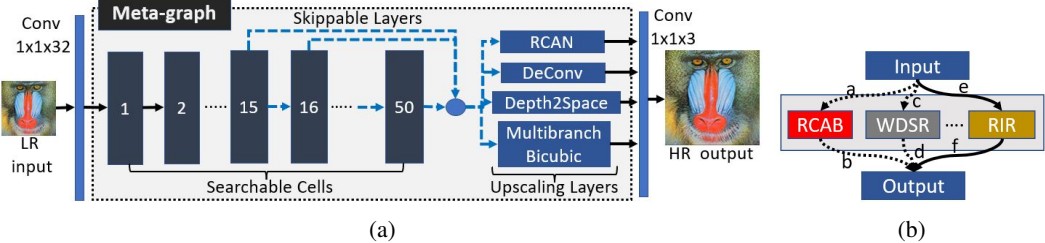

(a)                                                                        (b)

Figure 3: Meta-graph with (a) search space and (b) adaptive sub-graph cell architectures. Each cell is search-able and all layers after 15 are skippable. Four types of upscaling layers can be searched.

***Meta-Graph Design***. The first step in designing a NAS framework is the Meta-graph. This needs to be diverse enough to generate a wide range of efficient candidate architectures with competitive performance with state-of-the-art. In our case, we focus on designing a meta-graph with a variety of inference FPS with a good set of PSNR values, rather than developing a single novel cell architecture that beats all state-of-the-art SR models. We design our meta-graph as a cell-based architecture (as shown in Figure 3a). The input layer and the last output layer are kept constant, and the architecture between them consists of layers of cells followed by an upsampling layer. During the search phase, we sample paths within each searchable cell such that only one type of cell block is chosen at any one time. Existing works have proposed a wide variety of cell blocks which were demonstrably efficient and have many operations in common, and so we use them for our search space. Specifically, we use the cells from CARN (Ahn et al. (2018)), FALSR (Chu et al. (2021)), RCAB (Zhang et al. (2018)), AutoGAN (Fu et al. (2020)), WDSR (Yu et al. (2018)), ESRGAN (Wang et al. (2018a))

and MoreMNAS (Chu et al. (2020)) since they are well-known, efficient, self-contained blocks. Another large benefit is that all these cell blocks share many of the same operation types, making it possible for them to share weights. The most commonly shared operations between them are the convolutional and dense layers. As such, we set their parameter search spaces as: • Convolutional Filters - 1x1, 3x3, 5x5, • Convolutional Channels - 3, 16, 32, 64, • Dense - 64, 128, 256. Cell blocks for RCAB, AutoGAN, ESGRAN and FALSR also have attention and residual connections, which we can enable or disable as paths since we find that while their inclusion result in increased performance, it is not significant at the higher end models when considering the FPS increases.

*Layers*. The number of layers is one of the most important factors when determining the overall FPS and is therefore a very important search parameter. Additionally, different layer types perform distinctly with different layer numbers (Wang et al. (2018a); Ahn et al. (2018); Chu et al. (2021)). Therefore, we set the range of our number of layers of cells from between 15 to 50 with intervals/groups of 5 for every single cell type. Lastly, for our upscaling layer, we keep the choices between four types – RCAN (Zhang et al. (2018)), Deconvolutional (Dong et al. (2016)), Depth2Space (Bhardwaj et al. (2021)) and Multi-branch bicubic (Behjati et al. (2021)). We have a total of number of possible cell types of 12 when including blocks with and without attention and residual networks and a total number of parameter combinations of 3 * 4 * 3 = 36, giving us a total number of 12 * 36 = 432 cell combinations. With possible layer combinations of 3 to 10, we have a permutation formula $^{432}P_{10} + {}^{432}P_9 + {}^{432}P_8 + ...{}^{432}P_3$ times 4 for the number of possible upscaling layers, which gives us a total of $5.8e10^{23}$ possible networks.

*Sampling Paths*. Given such a large number of possible graphs, we need an efficient method for traversal. Fortunately, our hard constraint of Eq. 1 for FPS can play an important part in reducing the space here. We first start with setting the number of layers to the minimum possible. This is a tunable hyperparameter, and we find that below 15 convolutional layer blocks results in all sub-graphs with low PSNR value. Therefore, we set it to 15 and start random uniform sampling models without replacement. We profile them on the desired hardware to ensure the FPS is under the $F_{max}$. If not, it is permanently discarded. Once we have $N_l$ models, we move to the next layer limit and iterate until we have reached the last layer limit $I$ (50 in our current setting) which gives us a total of $N_l * I$ models. Note that for our scenario, we need multiple models across a wide FPS spectrum. In other words, we need multiple choices of models for the same dataset deployable for different hardware and levels of resource availability. Therefore, before starting the training we first determine suitable models by *binning* them. We select how many bins of FPS we need as $B$ with each bin having a window size of 200ms, and we bin our profiled models within each bin. We then train and evaluate each model within each bin for 5 epochs and evaluate them which gives us a sufficient estimate of final model performance. We now need to use an evaluation criteria to rank $N_b$ models per bin. For this method, we set a desired PSNR threshold for bin $b$ decided by the Pareto frontier. For example, from figure 8 we see that models on 1080Ti with 20 FPS needs to have at least 34 PSNR to be competitive, so we select that as the threshold. Then we take all models above that threshold and rank them based on the number of intersecting operations with others $\mu$ from Eq.1 We iterate this process until all bins $B$ have $N_b$ models. This way, we have a total of $N_b * B$ models for exploration. The parameters $N_l$, $N_b$ and $B$ are very important for determining the search and model efficiency for our framework and their choice is discussed later in Section 5. We throw out all other models which are not within the bins and redeploy the meta-graph with only the selected model operations. This reduces the number of models in the search space drastically and reduces weight sharing which further alleviates the co-sharing issue.

*Training*. After reducing the search space to $N_b * B$ models, we then train them till convergence. As mentioned above, we perform GAN training. Our predictor models are the searched models and for our discriminator we use the pre-trained VGG-16 model. We use the Adam optimizer with loss function for the generator defined as -

$$l_{gen} = 1e^2 * |x_{hr} - G(x_{lr})| + l_{vgg}(G(x_{lr}), x_{hr}) + (5e^{-3} - mean(D(G(x_{lr})))) \qquad (2)$$

where $x_{hr}$ and $x_{lr}$ are the HR and LR images respectively, $G$ is the generator network, $l_{vgg}$ is the VGG loss (Wang et al. (2018b)) and $D$ is the discriminator.

## 4.2 ADAPTIVE SUB-GRAPHS

Standard implementations employ static graphs with fixed inputs and outputs pre-set between operations and are not changed during forward passes. As a result, traditionally designed meta-graphs

cannot change operation sequences in real time. Instead, they have to reroute the batches among different model inputs for inference.

For our case, we have a single large model in the form of a meta-graph. It contains sub-graph architectures which share operations. In order to switch between sub-graphs, we can change the input-output paths for the operations. We take advantage of the non-static graph property to implement this. We first define the sub-graphs as a network with the nodes representing the operations and the input-output variables as the end-point for paths between them. We store each of the meta-graph operations as node objects with its input and output variable references within. During inference, we can change the input/output variables of one operation node to point to and from another node. This effectively decides the execution path of the input batch, thereby defining the sub-graph architecture itself. Since we have all the operations with their trained weights loaded in memory, we can call every available operation with our desired input values at any time. Thus, when we want to execute a specific sub-graph path, we start by first selecting the required operations from the list of all operation nodes. We then create the sub-graph architecture by routing the output variables (which are pointers to Tensors in memory) of one operation to the input of the next desired operation by changing the variable references. We can do this by applying the equal operation of the output variable of one operation node to the input variable of the next operation node, which essentially links them together into a direct execution path. Since we can set the output pointer variables to the input variables at any time between forward passes without disrupting the computation results, we can re-route paths at a per-batch granularity. Thus, by changing the pointer variables of input-outputs, we can select sub-graphs architectures as per our requirements. This allows us to adapt between sub-graphs in real-time without degrading QoS.

## 4.3 MODEL ADAPTATION POLICY

---

**Algorithm 1** Model Selection Policy

---

**Inputs:** Meta-graph after training $A^*$, CPU/GPU utilization % $\rho$ at time $c$, maximum allowed FPS $F_{max}$, empty profile table $T$, $\alpha_{sel}$ is the list of sub-graphs that meet the selection criteria in descending order.
**for** each $\alpha_i$ in $A^*$ **do**
    **for** each $\rho_j$ **do**
        $T_{i,j} = Profile(\alpha_i, \rho_j)$
    **end for**
**end for**
**while** $True$ **do**
    $\rho_c = CurrentResources()$, $\alpha_{sel} = []$
    **for** each $\alpha_i$ in $A^*$ **do**
        **if** $T_{i,c} \leq F_{max}$ **then**
            $\alpha_{sel}$ += $\alpha_i$
        **end if**
    **end for**
    $\alpha_{sel} = Sort(\alpha_{sel})$ based on max PSNR
    Switch to model $\alpha_{sel}[0]$
**end while**
return $None$

---

We design an adaptation policy to decide the model adaptation strategy in real time for a given resource availability. This is done in two steps. First, we generate a profiling table which contains the frame rate of each model under different resource availability (e.g., utilization) for a device. We profile each model using 20 image frames and use the average latency to infer frame rate. The profiling cost is $no.\ of\ models * resource\ availability\ granularity$. This step is a one-time offline effort and thus the profiling cost is not a major overhead. During the real-time SR task, we keep track of the resource availability of system via resource monitoring tools. Once we observe a change in resource availability, we adapt the model by using the search criteria $\alpha_{sel}$ where $fps(\alpha_{sel}) \leq F_{max}$ and sort the candidates based on their performance. We use the adaptive sub-graph mechanism to select the top performing sub-graph and switch to it in real time to maintain QoS. The algorithm is constantly running while the application is active. It does not output any specific variable, but constantly controls the switch between sub-graphs (see Alg. 1 for details).

## 5 EVALUATION

| Model | # of Params. | Chkpt. Size |
|---|---|---|
| Assembled | 11.3M | 88MB |
| *DySR* | 8.44M | 67MB |

| Hardware | FLOPS | Freq. |
|---|---|---|
| Snapdragon 855 | 899G | 2.8 GHz |
| Intel i5-560M | 388G | 2.9 GHz |
| Nvidia 1080Ti | 11.3T | 1.4 GHz |
| Nvidia A100 | 312T | 1.3 GHz |

Figure 5: Model and hardware device specification comparison. Checkpoint Size is the size of the file on disk of the trained weights stored by Pytorch via its checkpointing mechanism

### 5.1 TRAINING SETUP

We implement *DySR* using PyTorch and perform the search and training using 4 A100 GPUs, taking 21 GPU days to complete per run. During the deployment, we perform a quick profiling (e.g., a few steps of forward passes) on the target device (e.g. Snapdragon 855) to measure its inference latency and create a profiled database. We then change our adaptation based on the expected inference latency from this profile. For training the searched models till convergence, we use Div2K training dataset with 64x64 patch size, batch size of 16 and learning rate of 0.0043. For the search parameters, we use the values 15, 2 and 5 for $N_l$, $N_b$ and $B$ respectively. For the video super-resolution dataset, we train with Vimeo90k (Xue et al. (2019)). The LR images and frames were generated by bi-cubic downsampling them by a factor of 2 and 4. To generate the Pareto Optimality curve, we use the the models as shown in Figure 6a).

### 5.2 BASELINES AND PARAMETERS

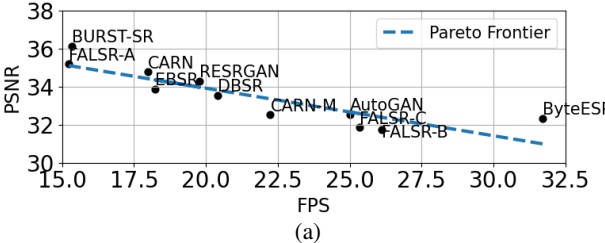

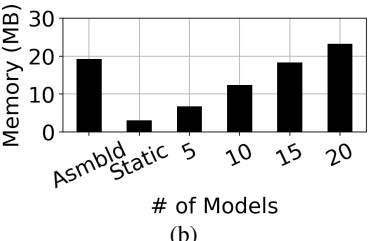

(a)

(b)

Figure 6: (a) State-of-the-art model's PSNR vs. FPS. Used to generate the Pareto curve determining the optimal trade-offs. Shown here for Div2K dataset run with 1080Ti for 2x scale. (b) Comparison of RAM consumption for *Assembled*, *Static* (CARN) and *DySR* with different numbers of sub-graphs generated by the $N_b$ and $B$ parameters. It is calculated as the sum of memory used for both model and activation (i.e., intermediate results).

We now describe our choice of baselines and search parameters. Since no other papers (to our knowledge) address this problem in the SR scope, we create our baseline model by combining all the models from Figure 6a (referred to as *Assembled*). It contains high-performing models over a wide FPS spectrum including the NTIRE 2022 Efficient Super-Resolution Challenge (Li et al. (2022)) winner ByteESR (Kong et al. (2022)). Our *DySR* should ideally generate similar models and so makes for a fair comparison. As mentioned above, the parameters $N_b$ and $B$ are very important in determining the quality of the models. Together they determine the total number of sub-graphs generated which in turn determines the amount of memory taken up by the meta-graph. It also indirectly determines how much weight is shared by the sub-graphs (more sub-graphs means more chances of operations being shared). In Figure 6b we show the memory consumed when executing *DySR*, Assembled and the single static CARN model after applying dynamic 8-bit quantization for all of them (Pagliari et al. (2018)). As we tune the $N_b$ and $B$, we get 5, 10, 15 and 20 models (no. of models = $N_b * B$). We see that even with 15 models, *DySR* has almost equal consumption to Assembled due to weight sharing among sub-graphs. With 5 models, *DySR* consumes around twice as much memory compared to the single model, illustrating the memory efficiency of our weight-shared NAS system. Since Assembled contains 10 models, we select 10 models for our *DySR* as well for fair comparison for the rest of the experiments. Table 4 shows that with equal number of models, our framework has less number of parameters and size on disk.

### 5.3 PARETO OPTIMALITY

We now test our searched and fully trained sub-graphs against state-of-the-art models. Figure 7 shows the PSNR vs. FPS performance of each of the 10 sub-networks in *DySR* for different datasets.

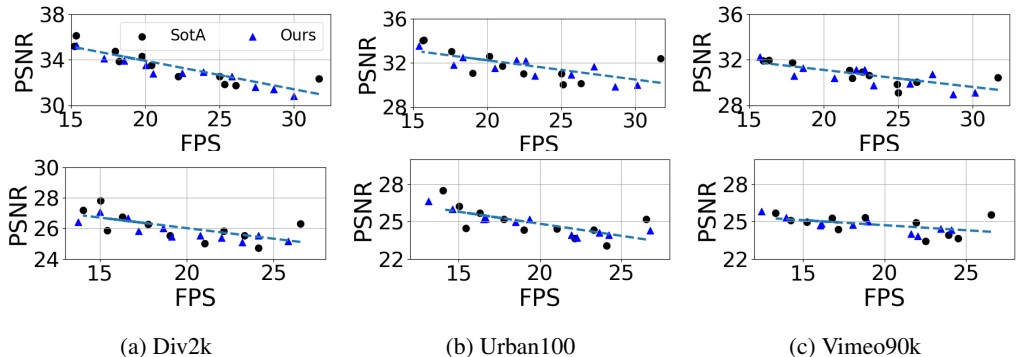

(a) Div2k      (b) Urban100      (c) Vimeo90k

Figure 7: PSNR vs. FPS for *DySR* generated sub-graphs for different datasets. Top row - 2X upscaling. Bottom row - 4X upscaling. Legend is applicable for all subfigures.

The sub-graphs are evaluated by passing the input through the execution path of the specific sub-graphs and their corresponding inference time measured. We observe here that across both image super-resolution and video super-resolution datasets for 2X and 4X upscaling, our sub-graphs achieve Pareto Optimality for the PSNR vs. FPS tradeoff created by the state-of-the-art models across the full FPS spectrum, demonstrating the efficiency of our search space.

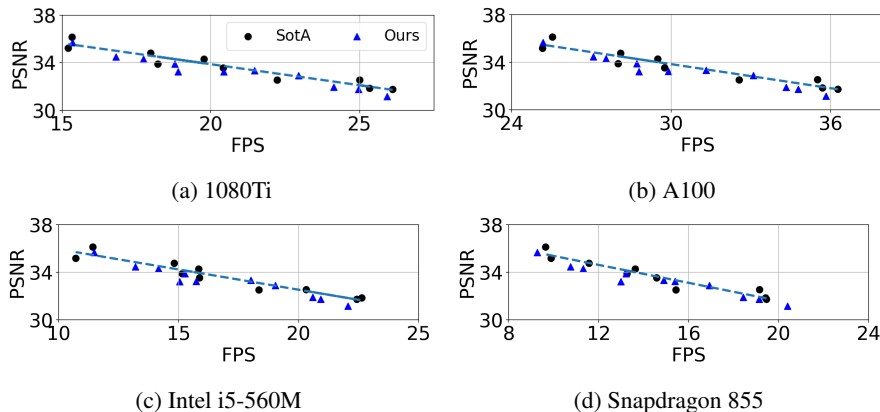

(a) 1080Ti      (b) A100

(c) Intel i5-560M      (d) Snapdragon 855

Figure 8: Pareto Optimality for different hardware. Generated by running *DySR* once and profiling FPS on 1080Ti to generate models between 15 and 30 FPS. While the FPS spectrum shifts across datasets, the Pareto Optimality is maintained. Legend is applicable for all subfigures.

We next evaluate our generated models against a diverse set of hardware, and the results are shown in Figure 8. We use a mobile CPU Snapdragon 855, a laptop CPU Intel i5-560M, a desktop grade GPU 1080Ti and a server-grade GPU A100 (Table 5). This figure shows the results for 2X scaling for the Div2k dataset (other datasets show similar results, so we eliminate them for lack of space). We search our models on the 1080Ti to be above 15 FPS. Therefore, the model's FPS shift for different hardware but still have models above 15 FPS (as was our search criteria). We see that across all these hardware, the Pareto Optimality of our generated sub-graphs are maintained, demonstrating that even with one search run we can generate efficient models for inference for such a wide variety of models. This property allows for model adaptivity since this allows the selection of models with any desired FPS for any hardware, and ensures close to the optimal HR output quality. Along with the ability to change models on-the-fly, *DySR* allows for optimal PSNR vs. FPS tradeoff for resource adaptivity as well, as will be demonstrated next.

## 5.4 DYNAMIC RESOURCE ADAPTIVITY

To demonstrate the resource adaptivity of our sub-graphs and model selection policy, we use three traces to generate real world resource fluctuation scenarios. We use a *bursty* trace to simulate extreme changes in resource availability, *stable* to illustrate a relatively small variation in resource availability and *random*. Since resource adaptivity has rarely been discussed in the SR context, we create two possible baselines. The *static* baseline contains a single model deployed at any one time from the list of state-of-the-art models used above for generating the Pareto Optimality curve. The

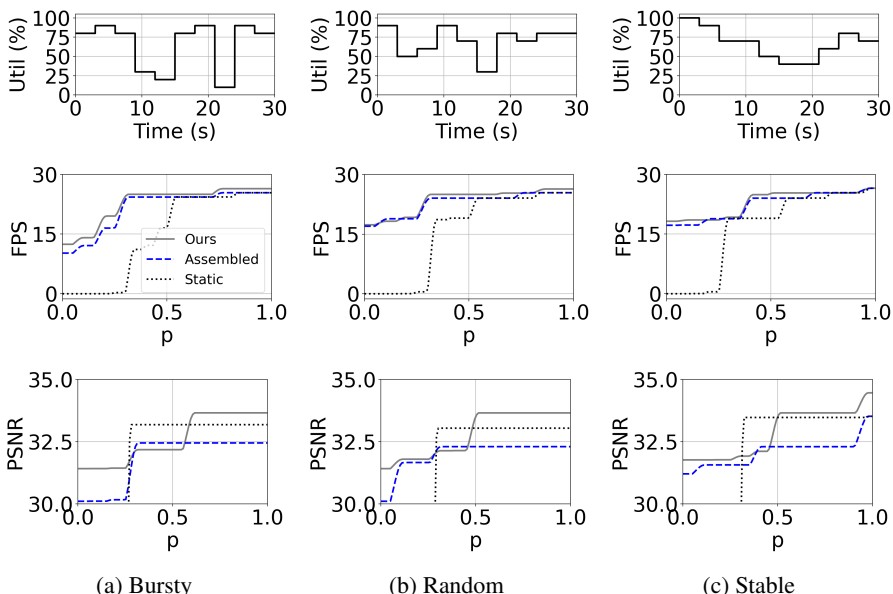

(a) Bursty         (b) Random         (c) Stable

Figure 9: CDF comparison between *DySR* and baselines for FPS and PSNR based on *bursty, random* and *stable* trace.

*Assembled* baseline contains all the models from the state-of-the-art list deployed at the same time to represent the scenario of keeping models pre-loaded. Based on available resources, we use our real-time swapping policy to choose the best model for both our *DySR* as well as the two baselines. Figure 9 shows the results of our experiment using the Div2k dataset on 1080Ti for 2x scaling. The top row shows the trace used, the middle row shows the CDF of the resulting FPS fluctuation over time, and the bottom row shows the the Cumulative Distribution Frequency (CDF) of the PSNR/FPS. It is the probability of the PSNR/FPS being less than a certain value at any given time. Higher y-axis values at lower $p$ values means better performance.

Here we observe that across all traces, *DySR* achieves overall higher values of PSNR compared to the baselines. The FPS for *Assembled* and *DySR* are somewhat similar due to both having little to no reloading overhead. However, *Assembled* performs worse for PSNR due to having less granularity of models available across the full FPS spectrum. For example, between 20 to 25 FPS in Figure 7 for Div2K with 2X upscaling, we see that our framework generates four models but the Assembled only has two. Therefore, Assembled has less options and tends to choose models with lower PSNR in order to be able to maintain its FPS. The *Static* models perform the worst due to having many interruption of services as expected. For the PSNR values, *Static* has a significant portion of its CDF distribution at 0 and therefore performs much worse than the others, especially at the lower $p$ values. Our framework outperforms all baselines by having the highest PSNR probabilities. Another observation is that the largest improvement in PSNR and FPS gained from *DySR* is for the *bursty* workload. This is mainly due to there being more model switches required here compared to the other two workloads. This amplifies the benefits derived from the model selection policy and granularity and thus a greater difference is observed. As the workloads grow more stable, the numbers vary less, illustrating that the more unstable a system the more it should benefit from our solutions. Based on all these observations, we conclude that *DySR* outperforms both baselines and can address the resource adaptivity issue.

## 6 CONCLUSION

In this paper, we focus on providing QoS for super resolution used in steaming services of mobile devices. We propose an algorithm and system co-design approach *DySR*, which employs an adaptive-aware one-shot NAS to generate models with Pareto-optimal performance, low switching overhead, and frame rate constraint that can be adapted in real-time via our low cost incremental adaptation method. *DySR* outperforms existing state-of-the-art models as well as the simple assemble method we explored by achieving the best performance while providing a steady frame rate throughput with a memory footprint that is 40% less compared to the 10-model Assembled baseline.

## 7 ACKNOWLEDGEMENTS

We are grateful to our anonymous reviewers for their valuable comments and suggestions that significantly improved the paper. This work is sponsored by National Science Foundation CAREER-2048044.

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

# A APPENDIX

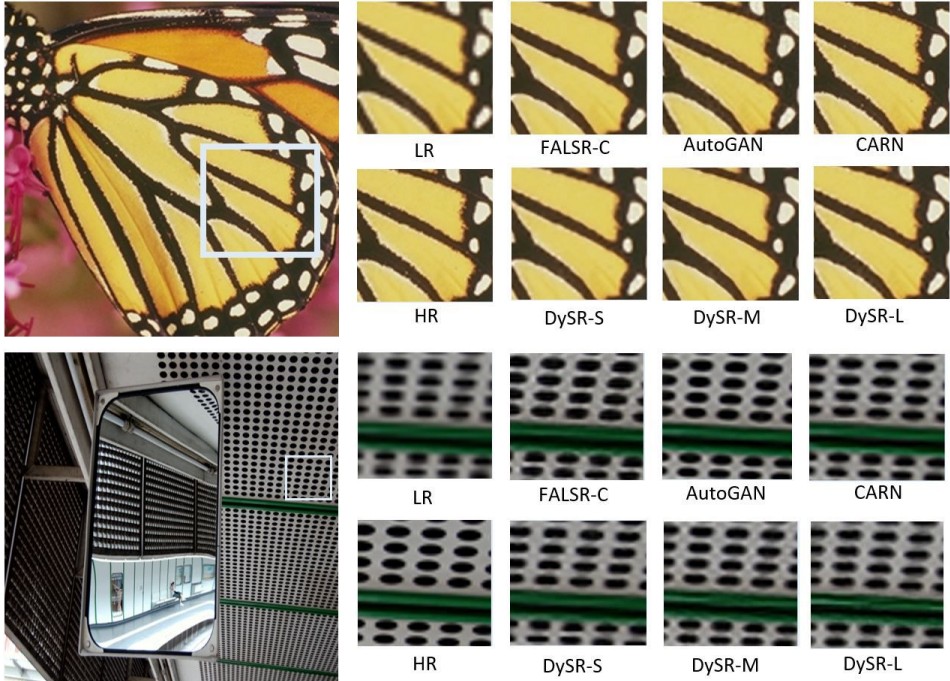

Figure 10: **Visual Comparison** - Comparing *DySR*'s HR output to other state-of-the-art models of similar sizes. For 4x upscaling, with samples from Set5 (top) and Urban100 (below)

## A.1 VISUAL COMPARISON

We compare our *DySR* outputs against other models in current literature with similar sizes. *DySR*-L,M,S represent models with 1.6, 0.74 and 0.38M parameters respectively. CARN Ahn et al. (2018), AutoGAN Fu et al. (2020) and FALSR-C Chu et al. (2021) have 1.5, 0.71 and 0.41M parameters. We observe that across both datasets, our HR outputs are quite similar to comparable state-of-the-art.

