# OpenReview forum: "DySR: Adaptive Super-Resolution via Algorithm and System Co-design"
_ICLR.cc/2023/Conference — ICLR 2023 poster_

### Official Review · Reviewer_bx3i · 2022-10-24

**Confidence:** 4
**Correctness:** 2
**Technical Novelty And Significance:** 3
**Empirical Novelty And Significance:** 3
**Recommendation:** 6

**Clarity, Quality, Novelty And Reproducibility:**

The document needs some editing. It would be difficult to reproduce their results from the description, and no source code is provided. Below I include some comments, questions, and suggestions:


*** Super resolution (SR) is a promising approach for improving the quality of low resolution steaming services on mobile devices-> streaming?

*** framerate -> frame rate?

*** define DySR


*** “This way, the amount of data moved during an adaption is minimized” There must be an objective function that is therefore optimized

*** Use proper parenthesis and improve the relationship between text and citation in the caption of Figure 1: “Uses FALSR-C, FALSR-B Chu et al. (2021), and CARN Ahn et al. (2018) models
respectively due to being designed specifically for the corresponding hardware.”

***” These results demonstrate that state-of-the-art SR models do not perform as expected under practical circumstances, and to maintain QoS (i.e., minimum
FPS), models need to be adapted according to the available resources.” How is this demonstrated?

*** advoid

*** “Therefore, before starting the training we first determine suitable models by binning them” -> Therefore, before starting the training, we first determine suitable models by binning them


*** What are the triangles and circles in Figure 8?

*** How is 15 defined as the minimum number of layers?

***Improve the sentence: “ Note that for our scenario, we need multiple models across a wide FPS spectrum such that for the same dataset so that we have choices on which models to deploy for different hardware and levels of resource availability.”

*** G in (2) uses one and two parameters. Please clarify.
***Define an output in Algorithm 1

*** What is rho in Algorithm 1? What is c?
*** One should assume that sort delivers ordered elements in alpha_sel in descending order. Please make it explicit in the document.


***Please fix “ We use the adaptive sub-graph mechanism to select the top performing sub-graph in real time and use adapt to it to maintain QoS (see Alg. 1 for details).”

***Define Eval and mu in (1)

*** In Figure 6 caption, sizeof(int8) is computer language dependent. Please explain in English.


*** “quantization for all of them Pagliari et al. (2018).” -> quantization for all of them (Pagliari et al., 2018). Review other reference calls.

*** In figure 7, the legend seems to refer exclusively to the subfigure in the upper left.

*** What is CDF in the Figure 9 caption?

*** Define Chkpt.Size in Figure 5.
*** How is the reduction of 40% in memory footprint estimated?

**Strength And Weaknesses:**

This study is the first paper addressing the problem of dynamically adapting a network architecture to available resources while providing constant throughput. In their approach, the results seem close to the Pareto frontier in the tradeoff between FPS and PSNR.

Please clarify how your 40%footprint reduction is achieved. The source code is not provided. The article requires some Editing.

**Summary Of The Paper:**

The authors present a model for adaptive super resolution on mobile devices. Their approach employs a one-shot neural architecture search to generate alternatives sharing weights. For inference, they introduce an incremental adaptation method. Their model aims to mantain a steady frame rate with a low memory footprint relative to baseline methods.

**Summary Of The Review:**

The article provides an interesting idea for adaptatively defining a super-resolution operation during streaming. However, the authors could improve the presentation. Also, verifying the source code for this method wiould be interesting.

The authors have addressed the reviewers comments. I raised my evaluation to "marginally above the acceptance threshold". Thanks

---

> ### Author Response · Authors · 2022-11-18
> **Response**
>
> Thank you for your valuable comments. We have revised the original draft and highlighted the changes according to your comments.
>
> 1. We will make our code open source upon acceptance. Here is a work-in-progress preliminary version of code: [Link](https://anonymous.4open.science/r/srnas-1001/) and we will add it in the revised draft.
>
> 2. 40% memory footprint reduction is derived from the results of Figure 6(b) when comparing 10-model DySR against the Assembled approach and we have clarified it in the revised draft’s Abstract.
>
> 3. DySR stands for Dynamic Super-Resolution and we have clarified it in the revised draft’s Abstract.
>
> 4. **“This way, the amount of data moved during an adaption is minimized” There must be an objective function that is therefore optimized**
>     - Here, we mean that since we do not need to load new models from the harddisk, there is no data movement overhead. We have clarified it in the revised draft’s Introduction.
>
> 5. **Use proper parenthesis and improve the relationship between text and citation in the caption of Figure 1: “Uses FALSR-C, FALSR-B Chu et al. (2021), and CARN Ahn et al. (2018) models respectively due to being designed specifically for the corresponding hardware.”.**
>   - FALSR-C and FALSR-B are from the same paper, so we cited them once. We have corrected it in the revised draft’s Figure 1 caption.
>
> 6. **These results demonstrate that state-of-the-art SR models do not perform as expected under practical circumstances, and to maintain QoS (i.e., minimum FPS), models need to be adapted according to the available resources.” How is this demonstrated?**
>     - In Figure 1, since the FPS drops at higher utilization, the QoS is impacted.  We have clarified it in the draft’s Section 3 paragraph 1.
>
> 7. **What are the triangles and circles in Figure 8?**
>    - Triangles are our DySR models and circles are State-of-the-art, We have added the legend.
>
> 8. **How is 15 defined as the minimum number of layers?**
>      - This is a tunable hyperparameter that serves as a control knob for users to control the search space and model performance. In our case, we find that below 15 convolutional layer blocks results in all sub-graphs with low PSNR value. Therefore, we set the limit to 15. We have clarified this in Section 4.1 Sampling Paths paragraph.
>
> 9. **G in (2) uses one and two parameters. Please clarify**
>      - lvgg(G(xlr,xhr)) should be lvgg(G(xlr),xhr) and we have corrected it in the revised draft’s Equation 2.
>
> 10. **Define an output in Algorithm 1**
>      - The algorithm is constantly running while the application is active. It does not output any specific variable, but constantly controls the switch between sub-graphs. We have clarified this by adding a “return None” in Algorithm 1.
>
> 11. **What is rho in Algorithm 1? What is c?**
>     - rho is the device utilization at time c. For example, we use a shell script to detect CPU utilization every 0.1 second and the value is used as rho. We have clarified this in the revised draft’s Algorithm 1.
>
> 12. **Define Eval and mu in (1)**
>     - Eval(A(α, D)) is the PSNR value of sub-graph α of the meta-graph A on dataset D. μ is the number of operations α has shared with all other sub-graphs in A except for itself. We have clarified this in the revised draft’s Section 4.1 - Adaptation-aware Sub-graph Selection paragraph.
>
> 13. **In Figure 6 caption, sizeof(int8) is computer language dependent. Please explain in English.**
>     -  The sizeof(int8) here is 4 bytes. We have clarified this in the revised draft in Figure 6's caption.
>
> 14. **One should assume that sort delivers ordered elements in alpha_sel in descending order. Please make it explicit in the document.**
>    - We have addressed this in the revised draft’s Algorithm 1.
>
> 15. **In figure 7, the legend seems to refer exclusively to the subfigure in the upper left.**
>    - It is for all graphs for Figures 7 and 8. We have added legend in Figure 8 and clarified it in their captions.
>
> 16. **What is CDF in the Figure 9 caption?**
>    - It is the Cumulative Distribution Frequency of the PSNR/FPS. It shows the probability of a certain PSNR/FPS at any time. We have clarified this in the revised draft’s Section 5.4 second to last paragraph.
>
> 17. **Define Chkpt.Size in Figure 5.**
>    - It is the size of the file on disk of the trained weights stored by PyTorch via its checkpointing mechanism. We have clarified this in the revised draft in Figure 5's caption.
>
> 18. Thank you for pointing out the grammatical and spelling errors. We have fixed them in the revised draft.

---

> ### Comment · Reviewer_bx3i · 2022-12-11
> **Opinion**
>
> I thank the reviewers for providing further insights and the authors for the clarification. After reading them all, I keep my rating.
>
> Thanks

---

### Official Review · Reviewer_9DkJ · 2022-10-25

**Confidence:** 4
**Correctness:** 3
**Technical Novelty And Significance:** 3
**Empirical Novelty And Significance:** 3
**Recommendation:** 6

**Clarity, Quality, Novelty And Reproducibility:**

The clarity of the paper is quite good. It has stated clearly the motivation, the challenge of current task, and the goal of the proposed method (system). While, the novelty is not quite related to developing new algorithm-based method, but to employing current solution to solve a practical problem. The originality of the work is high that there is no other method/manuscript that solves similar problems.



**Strength And Weaknesses:**

Strengths
1. This paper has clearly pointed out the strong motivation of the work and the necessity of getting a ad-hoc switchable super-resolution system that can automatically adjust the model based on the resources available in the system. I do believe the proposed method is useful in practical situations on mobile devices, especially for those low-end devices or embedded systems.
2. This paper has compared the proposed method with state of the arts baseline methods and outperform them in terms of the processing speed as well as the accuracy using Pareto curve to indicate the validity of performance.

Weaknesses
1.  In the algorithm, one of the most important input factors to decide sub-graph is the current available resources in the devices. However, i could not find the descriptions and explanations about how to to quantify the current resources, for example, you need to balance a bunch of parameters including memory, traffic, load on CPU /GPU , etc.. This is, I believe, the important starting point to decide whether your solution is applicable to mobile systems.
2. In this experiment, the training step is conducted on desktop-level system on A100 graphic cards. However, it is not quite clear about the deployment pipeline. If the model needs to be trained off-the-shelf and deployed on a mobile system or it should be trained on the target system otherwise the input hardware parameters are not available beforehand.



1. There are quite a number of typos and grammar errors:
(1). Abstract: Line2 steaming services -> streaming services
(2). Section 3 Motivation and Challenge: Line 8:  ..., see Figure 1 -> as shown in Figure 1. Two verbs in one sentence.

**Summary Of The Paper:**

This paper proposes a systematic solution for image super-resolution tasks. In order to provide good image/video quality under limited resources on mobile devices, the proposed method employs adaptive Neural Architecture Search (NAS) technique to produce model adaptation with minimum overhead. The proposed method aims to dynamically switch SR models that consistently adapts to the varying computer environments so as to balance the visual quality of super-resolution and the limited resource on the mobile devices. Comprehensive experiments show that the proposed method is able to provide steady throughput.


**Summary Of The Review:**

In summary, this paper is creating a new area in image super-resolution domain, which is quite practical and engineering-centric. I would like to suggest the authors to address the problems in the weaknesses mentioned above.

---

> ### Author Response · Authors · 2022-11-18
> **Response**
>
> Thank you for your valuable and encouraging comments.
>
> 1. Here we use the instant CPU or GPU utilization as an example of resource load measure and then adapt the model based on the profiled results (e.g., the most profitable model adaptation at each utilization level). We use tools such as “top”, “nvidia-smi” and Android Profiler to measure CPU/GPU utilization ρc = CurrentResources() every 0.1 seconds, and report it back to the program described in Algorithm 1. We can extend it to more resources such as memory, traffic.
>
> 2. We use A100 GPUs for training the model, but during the deployment, we perform a quick profiling (e.g., a few steps of forward passes) on the target device (e.g. Snapdragon 855) to measure its inference latency and create a profiled database. We then change our adaptation based on the expected inference latency from this profile. It is possible to do the training off site because our model is adaptive, which is one of the advantages of DySR. We have clarified this in the revised draft’s Training Setup Section and highlighted them.
>
> 3. Thank you for pointing them out. We have fixed the typos in the revised draft.

---

### Official Review · Reviewer_bgMe · 2022-10-25

**Confidence:** 4
**Clarity, Quality, Novelty And Reproducibility:** The paper is well written.
**Correctness:** 3
**Technical Novelty And Significance:** 2
**Empirical Novelty And Significance:** 3
**Recommendation:** 5

**Strength And Weaknesses:**

Strength

This paper aims to develops an image SR method on the QoS. The main goal is to propose an efficient method that can work well on the limited computing resources. The provided results demonstrate the effect of the proposed method to some extent.

A new meta-graph design is developed. In addition, introducing a NAS for the SR is interesting.

Weakness:

My major concern is experimental evaluation.

There are lots of methods that focus on developing efficient SR for mobile devices [e.g., NTIRE 2022 Challenge on Efficient Super-Resolution: Methods and Results, CVPRW 2022]. The paper does not compare with these methods. It is not clear whether the proposed method performs better or not.

In addition, no visual comparison is provided.

**Summary Of The Paper:**

This paper proposes a DySR method that maintains QoS while maximizing the model performance. The proposed method is mainly based on NAS. Experimental results show the effect of the DySR.


**Summary Of The Review:**

The experimental evaluations are not enough.

---

> ### Author Response · Authors · 2022-11-18
> **Response**
>
> Thank you for your valuable comments.
>
> For the NTIRE 2022 challenge, we did not include the comparison with it in the original draft because it was not available at the time of the submission. Thanks for pointing this out and we have added the comparison results with the winning ByteESR model from the 2022 NTIRE Challenge for Efficient in the revised draft, see Figures 6 and 7.
>
> We would like to emphasize that the focus of this work is not to develop a single model that achieves state-of-the-art performance. Instead, we aim at designing an adaptive method that can provide QoS in dynamic resource environments.
> In addition, thanks to the NAS method we used, any future novel architecture would help fine-tune the search space and thus help improve the adaptive model with a better Pareto-Frontier. E.g., the ByteESR model from the 2022 NTIRE Challenge has a highly efficient cell architecture that can be integrated into our search space. We plan to update the results with this new cell architecture in a later version of the paper.
>
> We have also added the Visual Comparison results in the Appendix of the revised draft, see Figure 10.

---

### Official Review · Reviewer_6CHN · 2022-10-27

**Confidence:** 4
**Correctness:** 3
**Technical Novelty And Significance:** 3
**Empirical Novelty And Significance:** 3
**Recommendation:** 8

**Clarity, Quality, Novelty And Reproducibility:**

Generating the dynamic switching graph based on a NAS process seems to be a novel contribution of the paper. The methodology has potential application across many domains where there is a quality of service requirement and varying computational resources.

A small issue to consider is the word "adaption".  The words "adaption" and "adaptation" mean the same thing (according to a quick search), but adaptation is more commonly used by print and book publishers.  Personally, I had not seen the word "adaption" before reading the paper and initially thought it was a mis-spelling.  However, it is apparently correct.  It does not need to be changed, but the authors might consider swapping "adaption" for "adaptation" to avoid other readers having the same impression.

**Strength And Weaknesses:**

The main strength of the paper is enabling real-time switching between different networks to maintain a certain quality of service under different computational environments. The optimization process used appears to identify useful sub-graphs of the overall network that are competitive with prior work of similar computational complexity.

The results are competitive with state of the art methods, with the addition of being able to swap models on the fly as computational resources change.



**Summary Of The Paper:**

The paper presents a method for adapting a single-image super-resolution [SR] deep network to different computational environments while maintaining a real-time quality of service. The method used is to identify sub-networks (graphs) within a larger network (graph) that are optimized to solving the SR problem but with varying levels of computational requirements. At run-time the whole network is loaded into memory, and then the computational paths used can be varied on the fly as computational resources change. Comparison with prior methods show the overall network and the subgraph network are competitive in their results with prior work with similar computational costs.

**Summary Of The Review:**

The concepts in the paper are of broader potential interest than super-resolution.  Within that narrow domain, the novel contribution is maintaining the minimum quality of service while adapting to computational resources.  Otherwise, the absolute performance is similar to existing methods at varying levels of computational complexity.  The method of identifying the relevant sub-graphs is of potential interest across other domains.  The paper is clearly written, and the comparisons with SOTA methods are appropriate.

---

> ### Author Response · Authors · 2022-11-18
> **Response**
>
> Thank you for your valuable and encouraging comments. We have changed all “adaption” to “adaptation” in the revised draft.

---

### Decision · Program_Chairs · 2023-01-20

**Decision:**

Accept: poster

**Justification For Why Not Higher Score:**

When compared with the methods in NTIRE 2022 challenge, the method does not show better performance.

**Justification For Why Not Lower Score:**

The authors addressed a new problem, and designed a novel method for handling this problem.

**Metareview: Summary, Strengths And Weaknesses:**

This work proposes a new framework that adapts a single-image super-resolution model to various environments (with different computation capacity), and meanwhile keep real-time quality of service. The framework identifies sub-models in the large whole model . During test-time the whole model is loaded, and then computational modules can be dynamically activated on the fly based on the changed computational resources. Most of the reviewers mentioned the paper is novel and useful. Reviewer 6CHN: Generating the dynamic switching graph based on a NAS process seems to be a novel contribution of the paper. Reviewer 9DkJ: Method is useful in practical situations on mobile devices, especially for those low-end devices or embedded systems. Reviewer bx3i: This study is the first paper addressing the problem of dynamically adapting a network architecture to available resources while providing constant throughput. These three reviewers are positive on this paper, and pointed out some issues, which are mainly about clarification of details and presentations and can be easily addressed.
The Reviewer bgMe recommends negative score, and his main concern is the lack of comparison with NTIRE 2022 challenge. Authors have added the comparisons to paper, and justified that the focus of their work is different from the challenge, i.e., they do not aim to develop a single model to get state-of-the-art performance. Instead, they aim to devise an adaptive method to provide QoS in dynamic resource environments. AC believes this justification is reasonable. Considering the efficacy and novelty of the method, and also the method can be quite useful for practical applications, AC recommends accept for this paper.

**Note From Pc:**

if the above contains the word "oral" or "spotlight" please see: "oral" presentation means -> notable-top-5% and "spotlight" means -> notable-top-25%. As stated in our emails, we are disassociating presentation type from AC recommendations

**Summary Of Ac-Reviewer Meeting:**

Reviewer 6CHN, 9DkJ, and bx3i are all positive on this paper, and only raised some issues about the clarification of details, which have been addressed. Reviewer bgMe did not reply about the discussion. His/her main concern is about the comparison, and authors have addressed the issue, and more importantly, the main focus of this paper is different from the suggested methods for comparison. The authors solved a new problem - devising an adaptive method to provide QoS in dynamic resource environments.